# Endodontic Management of Three-Rooted Mandibular Second Molar with Three Separate Orifices and Three Independent Portals of Exit: A Case Report

**DOI:** 10.3390/healthcare11111528

**Published:** 2023-05-24

**Authors:** Kapil R. Jadhav, Cinnamon L. VanPutte, Polymnia Tsotsis

**Affiliations:** Southern Illinois University School of Dental Medicine, 2800 College Ave, Alton, IL 62002, USA; cvanput@siue.edu (C.L.V.); ptsotsi@siue.edu (P.T.)

**Keywords:** mandibular, second molars, three roots, root canal treatment

## Abstract

Most mandibular second molars are usually found to have either one or two roots. However, mandibular second molars can also present with variations in the number of roots as well as differences in the morphology of their root canals. An 18-year-old male presented to the Department of Graduate Endodontics clinic with a morphologically variable mandibular second molar with three roots—two mesial and one distal. Two periapical radiographs were taken at different angles, revealing that there were three different canals in separate roots, each with independent portals of exit. This is a rare anatomical configuration. The success of endodontic treatment depends on accurate diagnosis, careful examination, identification of additional roots and canals, as well as detection of variations in root canal morphology. Failing to recognize these variations may lead to failures of root canal treatments and thus unsuccessful endodontic treatment.

## 1. Introduction

Endodontic treatment is employed when the pulp of a tooth has been either injured or exposed. This type of treatment preserves the natural tooth, thus avoiding future problems with occlusion or function. There are four primary objectives to endodontic treatment: (1) to remove any pathological bacteria and prevent their further infiltration into the root canal; (2) to obturate the apex of the root such that no fluids may enter or leave the root canal, which would allow for further bacterial growth or inflammation of the surrounding tissue; (3) to obturate the access cavity in the crown of the tooth such that no further bacteria may enter the pulp chamber; and (4) to allow for healing of any peri-apical tissue inflammation [1].

Given that these four objectives involve accessing the root canal system of a damaged tooth, successful endodontic practice requires a thorough understanding of both the tooth anatomy and the internal root canal. This, when combined with a correct diagnosis and the adequate cleaning and shaping of the root canal system, leads to a predictable and successful outcome of root canal treatment. The failure of a clinician to detect extra roots or canals in symptomatic teeth is a major reason for unsuccessful endodontic treatment [2].

Humans exhibit a wide variation in the anatomy of each tooth type regarding the number and shape of roots and root canals. This wide variation inspired the development of a root canal configuration classification system. Historically, the first attempt at a classification system dates to Hess and Zurcher (1925) [3]. In 1969, Weine et al. provided descriptions of the root canal system of teeth but included only single-rooted teeth; there was no inclusion of multirooted teeth or complex arrangements [4]. Then, in 1974, Vertucci et al. included complex configurations of the root canal system and classified the configurations into eight different types (Figure 1) [5]. More recently, Sert and Bayirli (2004) added further complex supplemental types [6]. However, it is the systems created by Weine et al. (1969) and Vertucci (1974) that are the most universally utilized [4,5].

Specifically, in mandibular second molars, the most common finding is two-rooted molars with two canals in the mesial root and one canal in the distal root [7]. Cooke and Cox (1979) were the first to describe single rooted mandibular second molars with C-shaped canals. These C-shaped canals consist of a continuous groove connecting two, three, or four canals [8]. Subsequently, using the clearing technique, Manning (1990) found that 22% of mandibular second molars had single roots, 76% had two roots, while only 2% had three roots. The shape of mandibular second molar canals is affected by both the age of the patient and their ancestry [9]. C-shaped canals were found to be more prevalent in Asians and people of Asian descent [10]. A study of both the external and the internal anatomy of 628 extracted mandibular first and second molars found that 84.1% of mandibular second molars had two separate roots, 15.9% had fused roots, and 1.5% had three roots [11]. Ferraz and Pécora (1993) used periapical radiographs to examine 328 patients and found that the incidence of three rooted mandibular second molars was 2.8% in patients of Asian descent, 1.8% in black individuals, and 1.7% in white individuals [12]. Kim et al. (2016) used CBCT analysis to examine the anatomy of root canals in Korean populations and found that of 1920 mandibular second molars arising from 960 patients, 41% had one root, 58% had two roots, and less than 1% had three roots [13].

The primary objective of any endodontic procedure is to adequately enlarge, shape, and disinfect pulpal spaces, followed by obturation of the root canals with an acceptable filling material. The evaluation of two or more periapical (PA) radiographs with multiple different horizontal angulations is important to reveal both variations in the anatomy as well as the complexity of the root canal system found in the treatment tooth. Cone-beam computed tomography (CBCT) is an important adjunct tool, which can be used preoperatively and intraoperatively in conjunction with radiographs to accurately assess the root canal system and aid in the management of complex endodontic cases [14].

Given this wide variation in the anatomy of the root canal system in mandibular second molars, clinicians should spend considerable time evaluating PA radiographs (when CBCTs are not available) to determine a specific diagnosis before embarking on root canal treatment. Additional radiographs, taken at different angles, may be needed for detection of extra roots and canals to confirm root canal variation in teeth [12,15,16]. Here, we describe the clinical management of a three-rooted mandibular molar with three different orifices, each with independent portals of exit.

## 2. Detailed Case Description

An 18-year-old male presented to our clinic for evaluation of pain in the lower right region of his mouth. The patient was referred to the Department of Graduate Endodontics from a private practice dentist for consultation and possible treatment. The patient complained of dull intermittent pain, occasional bleeding from the tooth, and slight discomfort with biting over the last few months.

Upon clinical examination, a large carious lesion filled with polypoid tissue measuring approximately 3 mm × 2 mm was observed in the mandibular right second molar (tooth #31) (Figure 2). The pulpal growth occupied the center of the carious cavity and was light pink in color (Figure 2a). In comparison with adjacent and contralateral teeth, the patient did not respond to cold testing of tooth #31 (application of a cotton pellet sprayed with Endo-Ice; Coltene/Whaledent Inc., Cuyahoga, OH, USA). There was slight tenderness to percussion, but no pain on palpation. Bitewing (BW) and PA radiographs were obtained as part of the radiographic examination (Figure 2b,d). Additional distal side PA radiographs with 20° horizontal angulation revealed two mesial and one distal root (Figure 2c).

Radiographic examination demonstrated a large carious lesion extending from the occlusal surface to the mesial pulp horn of the tooth along with widening of the periodontal ligament space on both the mesial and distal roots. The periodontal probing depths were measured to be within normal limits. There were no significant findings in the head and neck examination. The medical history was non-contributory, and the patient reported no known drug allergies. Based on clinical and radiographic findings, a pulpal diagnosis of necrotic pulp (NP) and a periodontal diagnosis of symptomatic apical periodontitis (SAP) were made for tooth #31.

Due to the complexity of the case, a treatment plan was made for non-surgical root canal treatment (NSRCT) utilizing two appointments. We explained to the patient that after the completion of our endodontic treatment, he would need definitive restoration, followed by crown lengthening and crown placement. After discussing treatment options and reviewing the patient’s medical history, informed consent was obtained for root canal treatment of one of his second mandibular molars (tooth #31). A pre-operative CBCT was recommended, but due to the patient’s financial constraints, a CBCT was not obtained. Additionally, the patient did not consent to pulpal tissue biopsy for histological diagnosis.

The root canal treatment was initiated after administering 2% Lidocaine with epinephrine (1:100,000) and application of a rubber dam under high magnification microscopy (Global Surgical Corp., St. Louis, MO, USA). The pulpal growth was excised after administering intrapulpal local anesthetic under pressure to control bleeding and to aid in visualization. The access cavity was prepared using a surgical round #4 carbide bur. The inflamed pulpal tissue was removed using an endo spoon excavator. The canal orifices were located using the DG 16 explorer (HuFriedyGroup, Chicago, IL, USA). There were three canals that were identified: two mesial side canals (mesiobuccal; MB and mesiolingual; ML) and one large distal side canal (D) (Figure 3a), as seen in Vertucci type VIII.

The canal orifices were enlarged using Gates-Glidden drills #2 and #3 (Figure 3a). The patency of the canals was verified using size #10-K hand files. The working lengths of the canals were determined using a Root ZX II electronic apex locator (J. Morita USA, Tustin, CA, USA). The canals were enlarged using the Vortex Blue rotary file system (Dentsply Tulsa Dental Specialties, Tulsa, OK, USA) according to manufacturer’s recommendations using a crown-down technique to the following sizes: MB and ML (30/06) and D (35/06). The root canal irrigation was accomplished using 5.25% sodium hypochlorite (NaOCl) solution with a side-vented 30 g irrigation needle. The canal patency was maintained using a #10-K file after use of each rotary file. The canals were dried with paper points and filled with calcium hydroxide as an intracanal medication between appointments. A sterile cotton pellet was placed in the pulp chamber and the tooth was temporized using Cavit g (3M ESPE, MN).

During the second appointment, the patient reported no symptoms. The calcium hydroxide was rinsed from the canals using irrigation with 5.25% sodium hypochlorite. The gutta-percha master cones were selected corresponding to the size of the root canal preparations and the length was verified using PA radiographs taken at two different angles (Figure 3b). A 17% EDTA solution was used as a final rinse with the use of the EndoActivator system (Dentsply Tulsa Dental Specialties, Tulsa, OK, USA). The canals were dried with paper points and then obturated with the corresponding sizes of the master cones using Endosequence BC sealer (Brasseler USA, Savannah, GA, USA) with the single cone and warm vertical condensation technique. Final PA radiographs were taken, and the root canal fill was verified (Figure 3c,d).

The access cavity was sealed with Cavit g and GIC Fuji II plus cement restoration. The patient was referred to his general dentist for the placement of the definitive restoration and crown. The recall visits were planned at 6- and 12-month intervals.

## 3. Discussion

The main objective of root canal treatment is the maximum debridement of all organic pulp tissue and any necrotic remnants prior to obturation. It is imperative that variations in anatomy and aberrant root canal morphologies are identified before beginning any root canal treatment. Mandibular molars have been known to present significant variations in root canal morphology. Anatomical variations in the external and internal anatomies of root canals can be found in all groups of teeth, individuals, and different ancestries. It is important to understand these variations in anatomy, but also to be aware of differences among various population groups [10].

Most dental anatomy and endodontic textbooks discuss only the normal root and root canal anatomy of mandibular second molars; however, as previously noted, there are wide variations in their root and root canal anatomy [17,18]. The lower mandibular second molar is usually found as having two roots, one mesial and one distal [19]. However, there can be up to two, three or four root canals in these teeth [20]. On the other hand, there have also been reports of single rooted mandibular second molars (those having conical roots with a large single canal [8]. It is important for clinicians to be aware of and have knowledge of these sorts of variations in root canal morphology. Based on the studies conducted by Manning (1990) and Ferraz and Pécora (1992), there is less than a 2% probability of having three rooted mandibular second molars [9,12]. In our patient’s case, understanding these anatomical differences was critical for the interpretation of the pre-operative radiographs and the determination of the master cone lengths. The successful endodontic treatment in this case was the result of acquiring two different angles of PA radiographs for tooth #31.

Exposed pulpal tissue, as observed our patient, is likely indicative of chronic hyperplastic pulpitis, commonly referred to as “pulp polyp”. However, this histopathological diagnosis was not confirmable due to lack of consent by the patient for a biopsy. Long-standing irritation and chronic infection can lead to the development of chronic hyperplastic pulpitis, which usually consists of granulation tissue, often covered by epithelium. The color of the polyp can range from cherry-red to opaque-white depending on the thickness and the degree of keratinization of the epithelium covering the surface of the polyp. Pulp polyps are commonly found in young children and adolescents but are rarely found in middle-aged individuals. They are typically asymptomatic, with the pulp testing normal upon thermal or electrical stimulation. Some patients may experience discomfort from pulp polyps due to irritation from food lodgment while eating. Pulp polyps can often mimic gingival polyps, but histological examination differentiates between them based on origin and type of tissue. Determining treatment and prognosis of teeth with pulp polyps depends on the degree of tooth destruction and tooth restorability [2,19].

Cone beam computed tomography (CBCT) is an important diagnostic aid for three-dimensional assessment of teeth to understand the complexities of their root canal systems. There are several manufacturers and models of CBCT, which are categorized based on field of view (FOV; limited-volume, medium-volume, or large-volume units). The scan volume determines the area included in the images. Small FOV CBCT scans are commonly used in endodontics to obtain high resolution images of teeth. These images improve diagnosis and management of complex endodontic cases. The downside of high-resolution medium or large CBCT images is exposure of patients to high levels of radiation. The principle of “as low as reasonably achievable” (ALARA) should be followed while choosing the most appropriate imaging protocol. Clinicians should consider total amounts of radiation doses over time when choosing between a higher number of intraoral radiographs and CBCTs. Since radiation doses are higher for CBCT studies, CBCTs should not be used for routine screening or endodontic diagnoses in symptom-free cases; CBCTs should be contingent upon individual patient needs and treatment complexity [14,21,22].

## 4. Conclusions

This case highlights the importance of a thorough knowledge of the root canal system in all different types of teeth and in all different populations of people. Root canal morphology can even vary between the same tooth type within an individual patient. Here, we described a successful endodontic treatment of a mandibular second molar with an unusual root canal anatomy: the presence of three separate roots. To successfully treat teeth with uncommon anatomies, multiple PA radiographs should be taken at different angles when CBCT is either not available or the patient does not give consent. Careful evaluation of multiple PA radiographs taken at different angles at the beginning of patient appointments, and during patient appointments is important for the identification of extra roots or aberrations in the number of canals. Knowing the patient’s specific root canal morphology can lead to successful root canal treatment of symptomatic teeth.

## Figures and Tables

**Figure 1 healthcare-11-01528-f001:**
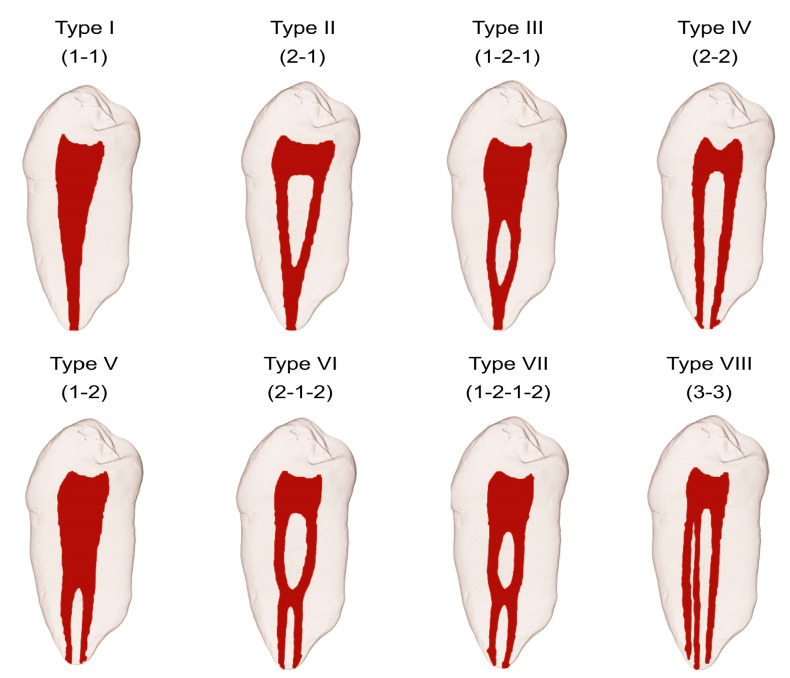
Vertucci root canal system. There are eight types (Types I–VIII) depending upon the arrangement of the canals. The canal arrangement varies on a spectrum ranging from having a single canal (Type I) to having three separate canals (Type VIII). Some intermediate arrangements include two separate canals that merge to form one canal (Type II), or one canal that, within the root, divides into two canals, but then merges into a single canal (Type III). Other intermediates have two individual canals (Type IV) or a single canal that divides into two separate canals each with their own apical foramina (Type V). Vertucci also described those that have either two separate canals (Type VI) or one canal (Type VII) that divide from the pulp chamber, rejoin within the canal, and then re-separate into two individual canals. (Used with permission of John Wiley and Sons: Lic. No. 5379651075635).

**Figure 2 healthcare-11-01528-f002:**
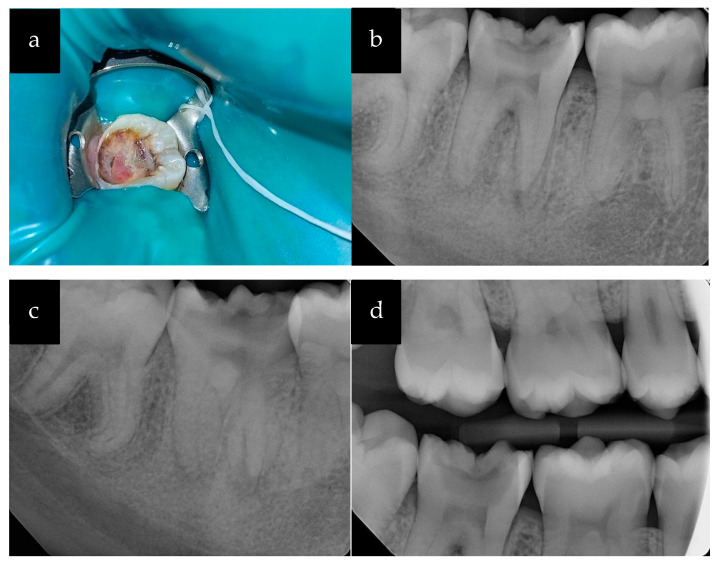
(**a**) Clinical photograph of tooth #31 with large carious cavity and significant pulp exposure; (**b**) Periapical radiograph; (**c**) Periapical radiograph with distal angulation; (**d**) Bitewing radiograph.

**Figure 3 healthcare-11-01528-f003:**
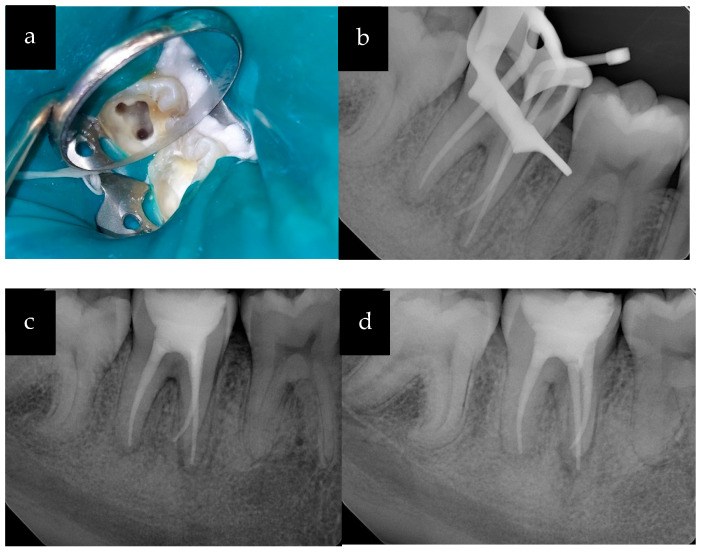
(**a**) Access cavity with three enlarged canal orifices; (**b**) PA radiograph showing the master cones; (**c**) PA radiograph: obturation and temporary restoration; (**d**) PA radiograph: obturation with distal angulation.

## Data Availability

No new data were created or analyzed in this study. Data sharing is not applicable to this article.

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
