# Peer review of "Endodontic Management of Three-Rooted Mandibular Second Molar with Three Separate Orifices and Three Independent Portals of Exit: A Case Report"

_healthcare, 2023, doi:10.3390/healthcare11111528_

Round 1

Reviewer 1 Report

The article is a case study for nonsurgical endodontic treatment of a three rooted second mandibular molar. It very aptly reviews literature present on the incidence and anatomical variations associated with mandibular molars. The article aims to describe management and diagnostic criteria for endodontic treatment. It highlights the importance of adequate diagnostic criteria for mandibular molars keeping in mind that vigilance needs to be maintained for less commonly occuring anatomic variations. The article aptly describes the diagnostic criteria for clinical diagnosis of endodontic lesions. The case report mentions that due to the unavailability of biopsy adequate diagnosis was not made but differential included symptomatic irreversible pulpitis and chronic hyperplastic pulpitis. However, chronic hyperplastic pulpitis is associated with non vital teeth but this tooth was symptomatic. I would also recommend a description of cold test, I am unclear on how to interpret positive response to endo ice. I want to know if the tooth responded, if yes then did it linger or not. The authors describe endodontic technique very thoroughly and it can be used as a teaching tool. I am curious as to the taper of the vortex blue file and whether that matters between three rooted teeth or not. There is no follow up of the study, I am curious about the prognosis of endo treatment for teeth with anatomical variations. Is there any data on this? I am also curious about what prompted the authors to take a PA with a different angulation, since it is not common clinical practice. Is there a clinical or radiographic rationale or do they recommend taking angulated PAs as a rule? The authors also mention using a microscope for treatment, I would like to know if better magnification would improve magnification of anatomy of pulpal floor thereby allowing for diagnosis of an additional canal. 

Reviewer 2 Report

Overall, interesting article on complexities of mandibular second pre-molars and follow up. 

Introduction should contain various imaging techniques that help in treatment planing these complex anatomies.  Additionally, importance of CBCT pre-operative and inter operative for treatment guidance should be added.

his case-report a pre-operative CBCT imaging would have strengthened this article.

Overall, well documented article, with minor grammatical errors that can addresed by proof reading

Minor grammar/syntax errors, will need proof reading.

Reviewer 3 Report

The following queries need to be answered and necessary amendments are to be done in the manuscript for considered for acceptance.

1. Specify the change in angulation for two periapical radiographs.

2. How the bitewing radiograph helps in identification of extra root as it does not cover the root surface, please clarify supported with picture.

3. Site evidence from published literature about the occurrence of two separate mesial roots. The evidence that you have provided includes radix paramolaris or radix entomolaris. Please clarify.

4. The sampling of tooth involves humans and therefore, informed consent from the patients must be supported. Ethical clearance should be supplemented. 

5. Despite the "ALARA" rule, the CBCT of the tooth should have been recommended.

6. Edit MD as ML

Round 2

Reviewer 1 Report

Thank you for considering my recommendations. 

Reviewer 2 Report

Thank you for your revisions.